# Ambulatory Care in Adult Congenital Heart Disease—Time for Change?

**DOI:** 10.3390/jcm11072058

**Published:** 2022-04-06

**Authors:** Louise Coats, Bill Chaudhry

**Affiliations:** 1Adult Congenital Heart Unit, Freeman Hospital, Newcastle upon Tyne Hospitals NHS Foundation Trust, Newcastle upon Tyne NE7 7DN, UK; 2Population Health Sciences Institute, Newcastle University, Newcastle upon Tyne NE2 4HH, UK; 3Bioscience Institute, Newcastle University, Newcastle upon Tyne NE2 4AX, UK; bill.chaudhry@newcastle.ac.uk

**Keywords:** adult congenital heart disease, autonomy, burden of care, self-management, health services

## Abstract

Background: The adult congenital heart disease (ACHD) population is growing in size and complexity. This study evaluates whether present ambulatory care adequately detects problems and considers costs. Methods: A UK single-centre study of clinic attendances amongst 100 ACHD patients (40.4 years, median ACHD AP class 2B) between 2014 and 2019 and the COVID-19 restrictions period (March 2020–July 2021). Results: Between 2014 and 2019, there were 575 appointments. Nonattendance was 10%; 15 patients recurrently nonattended. Eighty percent of appointments resulted in no decision other than continued review. Electrocardiograms and echocardiograms were frequent, but new findings were rare (5.1%, 4.0%). Decision-making was more common with the higher ACHD AP class and symptoms. Emergency admissions (*n* = 40) exceeded elective (*n* = 25), with over half following unremarkable clinic appointments. Distance travelled to the ACHD clinic was 14.9 km (1.6–265), resulting in 433–564 workdays lost. During COVID 19, there were 127 appointments (56% in-person, 41% telephone and 5% video). Decisions were made at 37% in-person and 19% virtual consultations. Nonattendance was 3.9%; there were eight emergency admissions. Conclusion: The main purpose of the ACHD clinic is surveillance. Presently, the clinic does not sufficiently predict or prevent emergency hospital admissions and is costly to patient and provider. COVID-19 has enforced different methods for delivering care that require further evaluation.

## 1. Introduction

The current prevalence of adult congenital heart disease (ACHD) in developed countries is over three per 1000 [1]. The growth of this group, due to improvements in medical and surgical care, is expected to continue until 2050 [2]. This is a heterogeneous group with different underlying conditions. The prospect of ventricular failure and residual structural problems increases with age and may be complicated by arrhythmia or sudden death. Extracardiac comorbidities are common and include mood disorders, thromboembolism and liver disease [3,4,5].

In contrast to other cardiovascular disorders or chronic conditions, those with ACHD generally remain within the specialist tertiary hospital setting throughout their lives. Investigations are complex and expensive, and the cost of health services is high and increasing [6,7]. Additionally, intermittent clinic nonattendance is frequent, and some become lost to follow up with worse clinical outcomes. Socioeconomically deprived groups are overrepresented in this group [8,9]. Many adults with CHD also report they are unaware of available healthcare services [10]. The costs and burden on the patient of outpatient healthcare are increasingly recognised in the wider healthcare setting [11].

The primary aim of this study was to explore how well our tertiary outpatient clinic detects clinically relevant problems in ACHD and consider the costs to the patient and health service provider. An additional aim was to define the level of nonattendance and any predictive factors. We also summarise the clinic activities during the COVID-19 pandemic when a hybrid approach of virtual and face to face consultations were arranged according to clinician perceived priority.

## 2. Materials and Methods

### 2.1. Study Population

One hundred patients attending the ACHD clinic at Freeman Hospital, Newcastle upon Tyne Hospitals NHS Foundation Trust between 1 October 2019 and 30 November 2019 were sampled by sequential hospital numbers. Demographic data (gender, age, ethnicity and postcode) were requested. Postcode was used to determine the index of multiple deprivation [12]. The diagnosis was determined from the electronic patient record and categorised according to the ACHD anatomical and physiological (ACHD AP) groupings and by survival probability [13,14].

### 2.2. Outpatient Appointments

High-level clinic data, including nonattendance, for the cohort of 100 patients was obtained for the period between 1 January 2014 and 30 December 2019. The electronic patient record was reviewed for each appointment. The letter reporting the consultation to the primary care practitioner was the primary source of information. The results of the investigations were determined from the clinician report rather than directly reviewing the source date. On 33 occasions, there was insufficient information available; these visits were excluded from the analysis.

To understand the impact of clinic attendance, we evaluated the decisions made, investigations performed and the clinical events occurring between clinic appointments. A clinical decision was defined as a change in management resulting from the assessment of the patient and included lifestyle and medication changes, multidisciplinary discussion for surgery, electrophysiology or intervention, referral to another specialist or shared decision-making with the patient against the recommended medical course of action.

### 2.3. Cost to the Patient

Distance and time travelled to the clinic was calculated using Google Maps (notional departure time 12 p.m. Tuesday) [15]. Workdays lost were calculated based on an average travel time less than 30 min requiring a half-day to attend and greater than 30 min requiring a full day to attend, an assumption based on the geography of the catchment area. Car travel time was used to calculate a best-case scenario for workdays lost, and public transport time was used to calculate a worst-case scenario. Clinic attendances to other departments within our hospital during the study period were also assessed.

### 2.4. COVID-19

Ambulatory care provision to the same cohort of patients, including nonattendance, during the COVID-19 pandemic (23 March 2020, 1st UK Lockdown to 19 July 2021, restrictions released) was also reviewed.

### 2.5. Patient Public Involvement

Prior to embarking on this study, a questionnaire was carried out in clinic to understand what patients felt about this topic area. We asked 57 adult patients why others may fail to attend appointments, 56% cited patient factors, most frequently that patients may feel too well or alternatively be too worried or anxious to attend. They also raised the issues of childcare and physical difficulty attending appointments. Seventy-seven percent felt this was or may be important to research. Whilst asking those who do attend this question may give a skewed response, it reinforced the authors belief that was a topic area that it was important to study. Additionally, our formal patient public involvement group expressed strong interest in this subject, discussing how outpatient clinics had remained unchanged for over 20 years.

### 2.6. Statistical Analysis

Continuous variables are presented as median and range due to non-normal data distribution. Categorical variables are presented as number and percentage. Comparisons between groups uses the Kruskal–Wallis test or chi-squared test, depending on the data type. Relative risk was calculated by dividing the probability of a decision occurring in those with the factor compared to those without. Significance was implied with a two-tailed *p*-value <0.05. Data were analysed in SPSS v.24. A sample size of 100 patients, identified by hospital number to minimise the selection bias, seen over 5 years was decided upon in advance, as we expected this size cohort to adequately capture the range of diagnoses and functional statuses seen in the clinic with a frequency of events (e.g., nonattendance, hospital admissions) sufficient to investigate the study aims.

## 3. Results

### 3.1. Study Population (n = 100)

The median age of the 100 patients studied was 40.4 years (range 29.7–75.8); 67% first presented in the neonatal period or during early childhood and, thus, had significant experience with the outpatient clinic through their life, while 95% had moderate or severe anatomical complexity, and 58% had minimal functional impairment (ACHD physiological classes A/B) (Figure 1a). Approximately half had Tetralogy of Fallot or valvular disease (Table 1). Those with pacemakers or defibrillators (16%) were reviewed by the cardiac rhythm management service during the appointment. This group had greater anatomical and physiological complexity (ACHD AP classes 2C–3D).

### 3.2. Attendance at Clinic

The lifetime clinic attendance of this group was 3195 years, with the study period representing just under one-sixth of that. Most patients attended clinic annually (Figure 1b); the frequency of attendances increased with the advancing ACHD AP class (Figure 2). Nonattendance accounted for one in 10 appointments. Thirty-five patients missed at least one appointment with 15 individuals recurrently nonattending (Figure 3).

Sporadic or recurrent nonattendance was unrelated to sex, age, diagnosis, ACHD AP group or socioeconomic group (Appendix A). Unlike others, we found no relationship between clinic nonattendance and emergency hospital admissions [16].

### 3.3. The Outpatient Clinic Visit (n = 575)

Eighty percent (459/575) of clinic appointments resulted in no clinical decision other than continuing review. Of these, 94% occurred in the context of surveillance, while 6% occurred whilst waiting for previously requested intervention, investigation or external opinion (Figure 4). We describe these as ‘monitoring’ appointments. During the five-year period, 53 patients had only monitoring appointments. Clinical decisions (20%, 126/575) were most commonly medication-related or a decision to discuss interventions at the multidisciplinary meeting. Decision-making appointments were more common in those with a higher ACHD AP class but not always related to the anatomical complexity of heart disease alone (Appendix A) and in the context of new symptoms, physical, ECG or echo finding (Table 2).

Electrocardiograms (ECG) and echocardiograms were frequently performed; however, new findings were rare (5.1%, 23/453 and 4.0%, 21/525, respectively) and more common with symptoms (*p* = 0.003), particularly new symptoms (*p* < 0.001). Twenty-two ECG findings were related to rhythm change and one to ST changes. Sixteen echocardiographic findings were related to valve disease progression, predominantly obstruction, four to change in ventricular function and one noted a small pericardial effusion. Where new physical, ECG or Echo findings occurred in the absence of symptoms, 10/17 were due to progress in valve disease, six to mild fluid overload and one due to an asymptomatic atrial arrhythmia. The median ACHD AP class of these individuals was 2C.

Additional investigations were requested at 28% of the appointments and usually required separate attendance (Appendix A). Cross-sectional imaging was most frequently performed in a tetralogy of Fallot (18/54) and aortic coarctation (16/54); two-thirds of 24-h blood pressure studies (8/12) were requested in aortic coarctation. Fontan patients received blood tests and gastrointestinal investigations (9/10 requests). During the study, 19 patients had 25 planned admissions for catheters, electrophysiological studies or surgery (Figure 5).

Sixteen patients were urgently admitted to hospital 40 times. Arrhythmia and heart failure were the leading causes (Figure 5). Over half (21/40) occurred following ‘monitoring’ appointments, where no notable ECG or echo findings were recorded. Furthermore, the clinical examination was unremarkable in all but one with a marginally elevated jugular venous pressure. Emergency presentations occurred across all diagnostic groups and ACHD AP classes. No relationship was found between emergency admissions and clinic attendance. In most cases (23/40), admissions were to hospitals not specialised in the care of ACHD. One patient presented overseas.

### 3.4. Cost of the ACHD Clinic

In our centre, the ACHD clinic is typically 4 h long and increasing in frequency alongside ACHD population growth (currently 10 clinics/week for a denominator population of 2.9 million). During the study, a typical clinic required three rooms managed by a clinic nurse and healthcare assistant. Consultations are provided by an ACHD specialist clinician (sometimes supervising a trainee), an echocardiographer and a specialist nurse. In addition, physiologists perform routine ECGs.

The clinic is held in a regional cardiothoracic centre in the outskirts of a major UK city. The median distance travelled to the clinic was 14.9 km (1.6–265), with time taken by car of 32 min (6–300) (corresponding public transport time 78 min (14–396)). Costs of travelling by car with parking or public transport were similar and in the region of £6 per clinic visit, although, for those on low incomes, costs may be reimbursed. Seventy-seven patients had a median of four (range 1–82) additional attendances to other clinics at our centre during the study period (Table 3).

Thus, most patients spent a median of £30 each year on clinic attendance. Additionally, between 433 and 564 workdays were estimated to have been lost through attendance at the ACHD clinic alone. These costs do not include attendance at the general practitioner or regional hospitals, which is also known to be high in the ACHD population. Importantly, nonattendance and recurrent nonattendance were not associated with distance to travel. Qualitative work suggests that, although adults with CHD find the clinic positively confirming, it is also a source of marked anxiety [17]. The emotional impact of clinic attendance could not be quantified here.

### 3.5. COVID-19 Pandemic

During the UK COVID-19 pandemic restrictions (23 March 2020–19 July 2021), 80/100 patients had 127 appointments (1.2 appointments/patient/year). One patient died prior to this period (AHA AP class 3C) following complicated high-risk surgery. Appointments were triaged in advance, and 54% were carried out in-person, 41% telephone and 5% video. Decisions were made at 25/67 (37%) in-person appointments and at 11/58 (19.0%) virtual consultations. There were six nonattendances: two in-person and four telephone. Nineteen patients remained unseen at the study close: seven were not due a follow-up, one was discharged to the local team, ten appointments were postponed due to clinic cancellation, and in one case, there was a failure to make a follow-up appointment. There were six elective admissions and eight emergency admissions.

## 4. Discussion

This study addresses the role of the outpatient clinic for the ACHD patient group and indicates a high cost of care delivery both to the patient and the service provider but a limited ability to detect and prevent relevant clinical problems. It confirms an emerging view that the traditional model of healthcare delivery by the outpatient clinic is of limited value [11] and raises concern regarding the perpetuation of this model of ambulatory care in ACHD that is reinforced by the ring fencing of expertise within specialist centres.

The population reported in this paper represent a white Caucasian group from a socioeconomically deprived region of the United Kingdom [12]. Whilst we believe this cohort to be representative of our outpatient population, it may not represent all ACHD clinic populations and is not representative of the whole ACHD population. Other groups have found that those with mild disease complexity, of male gender and with no prior procedures are less likely to attend clinic [18,19]. Mild conditions (which may tend to have a male predisposition, e.g., bicuspid aortic valve) were underrepresented in our cohort. Our results are therefore all the more concerning, as high levels of nonattendance still occurred despite greater levels of disease complexity. We also found no relationship between the socioeconomic group and clinic attendance: other groups, with more diverse populations, have found this to be a risk factor alongside ethnicity but further work is required to understand whether this is causal or confounding [8]. Whilst older than our overall clinic demographic, the cohort in this study reflect an age group more likely to experience CHD related complications and, thus, a more pertinent group in which to examine the study aims. Evidence suggests that nonattendance is even greater in younger adults with CHD, which is assumed due to failures in transition [8]. However, considering our findings, we should also ask how the outpatient model of healthcare delivery and its perceived effectiveness is valued by this group who will generally be in better health and have many other competing activities.

It is clear that the central purpose of the ACHD outpatient clinic is to provide surveillance to detect complications, e.g., valvar dysfunction, arrhythmia, progressive cyanosis or heart failure and initiate intervention that is beneficial. However, it is notable that the evidence base for this is mostly consensus-driven [20]. In our cohort, surveillance through the outpatient clinic had limited success, as multiple emergency admissions to non-specialist hospitals still occurred, most frequently for arrhythmia and heart failure. This suggests that screening in the ACHD clinic is ineffective. Moreover, it was delivered at significant and disproportionate cost to those individuals who lived further away or who had more advanced disease and were required to attend more often. Similarly, routine investigations (ECG and echocardiography) were frequently and universally performed at high cost to the patient and health service provider, despite new findings that led to clinical decisions being rare. Importantly, these investigations failed to predict future emergency presentations. Many ECG and echocardiogram parameters suggested to prognosticate in ACHD have limited value when assessing the individual in the outpatient setting, as population variability and their surrogate nature confounds their ability to discriminate well for clinical outcomes.

It is only in recent times, and with the growth of the ACHD population, that congenital heart disease has been recognised to be a chronic condition with different disease-based trajectories [21]. As with other chronic diseases, the trajectories are also influenced by socioeconomic circumstances [22]. The vision of the World Health Organisation for chronic conditions is that optimal care delivery is achieved through an integrated care approach centred on the patient and family, supported by the community, with expeditious utilisation of healthcare services [23]. This is consistent with the chronic care model, a framework that has been used successfully to develop new approaches to ambulatory care delivery for other chronic conditions [24]. Developing cost-effective surveillance for adults with CHD, which places the patient at the centre of their care, requires clear definitions of diagnostic groups according to morphology and surgical history with an understanding not only of long-term clinical risks in each group but how all stakeholders can best integrate to optimise health. Whilst specialist clinics (e.g., Fontan) are developing to try and rationalise clinic attendance for those with complex needs, this approach still fails to address the burden for the individual, such as travel, and neglects to properly integrate with other care providers and cross-sector organisations.

Along with a change in philosophy, screening modalities must be evaluated with multicentre studies to prove the efficacy. The “pick-up rate” of a test that is deemed acceptable must depend on a number of factors, including the intrusiveness of the test, the financial cost of the test, whether an alternative exists that is more specific and whether positive findings lead to change or continued observation. The most frequent clinic ECG finding resulting in a clinical decision and the most common reason for emergency attendance outside of clinic was a change in cardiac rhythm. Remote monitoring (Alivecor Kardia Mobile, Cardiacsense) devices may offer better tools to detect rhythm change than the temporally limited nature of clinic detection or Holter monitoring [25]. Similarly, given the difficulty in assessing function of complex cardiac morphologies, biomarkers such as NT-BNP and serial weight assessment may screen and monitor therapy more effectively for heart failure than echocardiography [26,27]. New technologies will further expand opportunities for mobile health [28], in the same way that home and online blood pressure monitoring has proven to achieve better hypertension control than traditional approaches [29]. Similarly, earlier detection of infective endocarditis or intracardiac thrombosis could be supported by patient and primary care education with community-based CRP and blood cultures or D-dimer measurements and enabling direct referral pathways. Cross-sectional imaging is increasingly the preferred modality to monitor for surgical complication, determine the timing of the intervention and plan the treatment. Echocardiography, whilst valuable as a bedside test, has a high resource implication in the outpatient setting and is arguably better suited to symptomatic and emergency investigation rather than surveillance in an asymptomatic individual, perhaps except for valvular heart disease, where reintervention is anticipated.

As with many other chronic conditions, defining better the role of the primary care practitioner in surveillance is key. At present, adults with CHD have high levels of attendance at primary care, but whilst ambulatory care is regarded to be the remit of the ACHD specialist. this presently translates to worse clinical outcomes [30]. Specialist nurses could provide the bridge between self-management, surveillance in primary care and the specialist centre in a manner that has already been achieved for those with acquired heart failure and provide, arguably more effectively, the additional roles of a clinic, including education and psychosocial support [31]. Understanding which aspects adults with CHD feel comfortable managing, where support can be given and determining appropriate thresholds and pathways for direct self- and primary care referral is a vital area for future research. A realistic view about the trajectory of health is also required. Early discussions regarding eventual transplant or repeat intervention should further inform and facilitate a personalised approach to self-management versus hospital attendance for ambulatory care.

Much data exists in our electronic healthcare system that can direct us but at present it is not collected or analysed from this perspective, rather, the focus remains on post-surgical mortality. The recently described ACHD AP classification provides a promising framework to stratify alternative models of care delivery and inform artificial intelligence interrogation of big datasets to predict risk in ACHD [13]. However, thought needs to be given to whether disease-based risk factors, that have conventionally been collected through the outpatient clinic model, are the most important or only drivers of the clinical outcome [32].

During the COVID-19 pandemic, different strategies for ambulatory care were necessarily tested. Whilst the design of this study does not enable conclusions to be drawn regarding the superiority or inferiority of these approaches, it does demonstrate that care can be provided in different ways, with good participation and incorporation of the patient into surveillance of their condition. The calculation of work days lost is also not a validated method, rather a pragmatic measure, with assumptions based on our experience of our local geography, to indicate that attending clinic has implications for patient’s lives above cost alone. This study is further limited by its recruitment strategy and retrospective nature. Patients who died or were transplanted in the five-year study window, as well as those already lost to follow-up, would have necessarily been excluded. Arguably, those with advanced disease may benefit more from face-to-face outpatient follow-up; however, it should also be acknowledged that this sicker group also has the highest burden of care with this model of care delivery. Whilst those lost-to follow up may present late with complications that could have been addressed earlier if they had remained under follow-up, the events leading to loss to follow up particularly in later life remain ill-defined and require further prospective study. Improvements in transition from paediatric services are rightly targeted, but events can also occur at other times during the life course that result in a patient losing contact with clinical services.

The present study raises concerns regarding the ability of the traditional outpatient clinic model of care to satisfactorily detect clinically relevant problems in adults with congenital heart disease. It is neither personalised nor minimally invasive and delivered at substantial cost to both the patient and healthcare provider. With growing evidence to support alternative methods of care for chronic disease, which empower patients whilst minimising the burden of care, we must ask ourselves why we, in ACHD, persist with a rigid outpatient clinic model of ambulatory care.

## Figures and Tables

**Figure 1 jcm-11-02058-f001:**
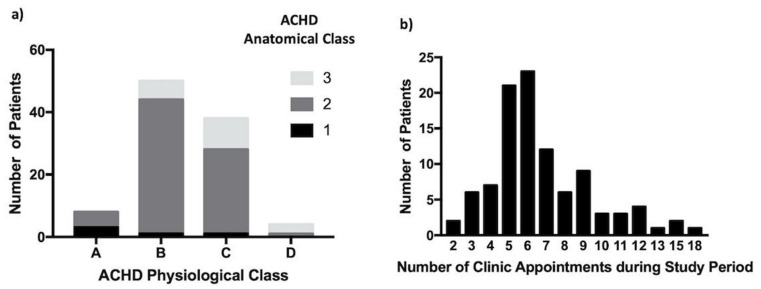
(**a**) ACHD AP class of the study cohort and (**b**) number of clinic appointments attended during the study period by the study cohort.

**Figure 2 jcm-11-02058-f002:**
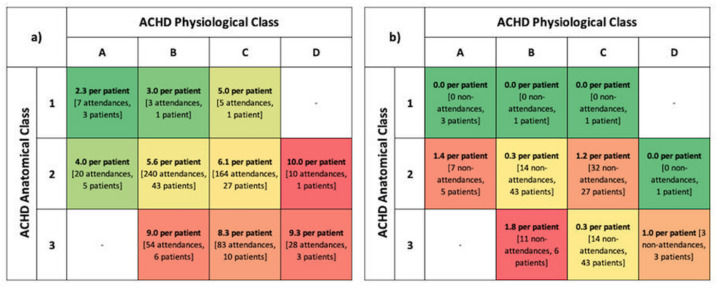
(**a**) Mean attendances and (**b**) nonattendances per patient during study period according to ACHD AP class.

**Figure 3 jcm-11-02058-f003:**
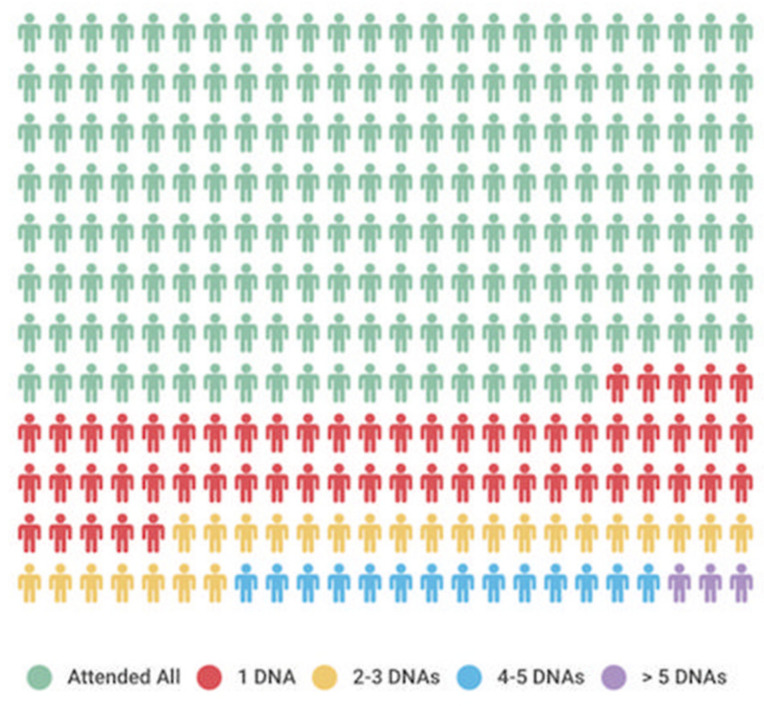
Frequency of nonattendance amongst the study population. DNA: Did not attend.

**Figure 4 jcm-11-02058-f004:**
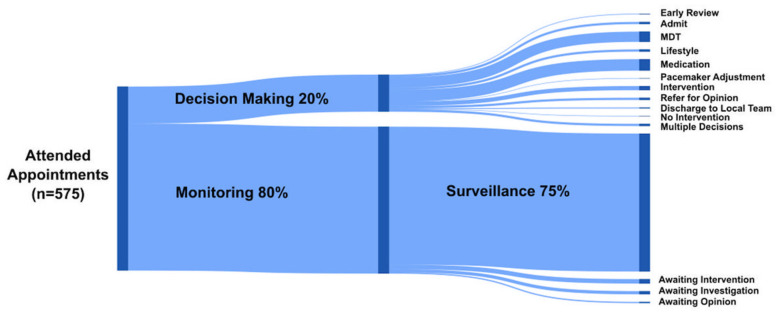
Proportion of clinic appointments during the study period resulting in clinical decision-making or otherwise defined as holding appointments.

**Figure 5 jcm-11-02058-f005:**
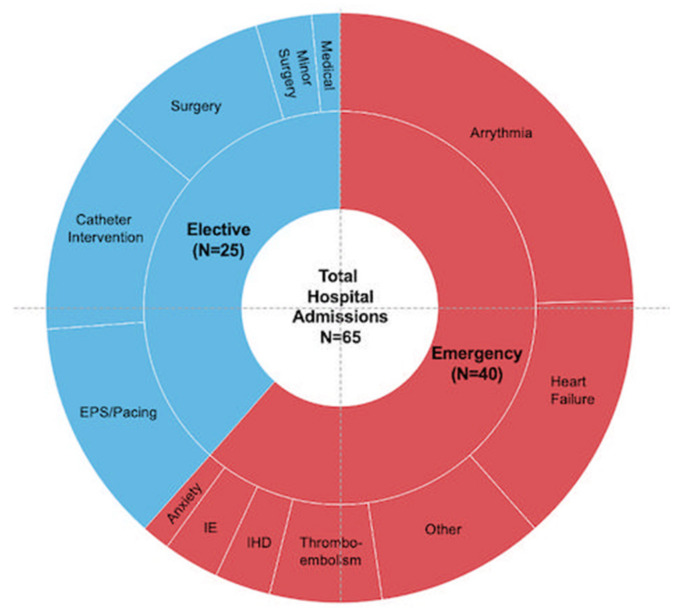
Reasons for hospital admissions. (IHD: Ischaemic Heart Disease; IE: Infective Endocarditis; EPS: Electrophysiology Study/Ablation).

**Table 1 jcm-11-02058-t001:** Study Population (n = 100) (AVSD: atrioventricular septal defect, ASD: atrial septal defect, VSD: ventricular septal defect and TGA: transposition of the great arteries).

Demographic	Median (Range)	N (%)
Male		54 (54%)
Age at 2019 Appointment	40.4 years (29.7–75.8)	
White British		100 (100%)
Index of Multiple Deprivation Decile * [12]	4 (1–10)	
Diagnostic Group [16]		
Tetralogy of Fallot		27 (27%)
Valvular Disease		22 (22%)
Aortic Coarctation		16 (16%)
AVSD		10 (10%)
Systemic Right Ventricle		8 (9%)
Fontan		4 (4%)
Complex Congenital ^#^		4 (4%)
ASD		3 (3%)
VSD		3 (3%)
Ebstein Anomaly		2 (2%)
TGA Arterial Switch		1 (1%)

* Decile 1 is most deprived, and decile 10 is the least deprived ^#^ Two patients had pulmonary atresia with a ventricular septal defect, and both had undergone biventricular repair, one following unifocalisation of the major aortopulmonary collaterals and the others following shunt surgery. One patient had pulmonary atresia with intact ventricular septum, and one had tricuspid and pulmonary atresia. Both were palliated with shunts alone.

**Table 2 jcm-11-02058-t002:** Relative risks of a decision being made in the context of different appointment factors (ECG: electrocardiogram; Echo: echocardiogram).

Variable	Relative Risk of Decision Being Made	95% CI	*p* Value (Fisher’s Exact)
Symptoms	2.446	2.067–2.894	<0.001
New Symptoms	4.294	3.056–6.032	<0.001
New Physical Finding	10.288	3.743–28.277	<0.001
Post-operative/obstetric review	2.793	1.373–5.683	0.006
New ECG or Echo Finding	3.957	2.239–6.994	<0.001

**Table 3 jcm-11-02058-t003:** Additional outpatient attendances during the study period (ACHD: Adult congenital heart disease).

Clinic Attended	Total Appointments	Number of Patients	Appointments per Patient (Median and Range)
Non ACHD cardiology	36	9	4 (1–11)
Cardiac surgery	31	12	2 (1–6)
Pre-assessment	37	32	1 (1–2)
Dental	70	28	1 (1–11)
Obstetrics or Foetal Medicine	114	9	13 (1–25)
Other specialities	436	59	4 (1–75)
Physiotherapy	9	2	-
All	733	77	4 (1–82)

## Data Availability

Data is available from the authors on request.

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
