# Peer review of "Ambulatory Care in Adult Congenital Heart Disease—Time for Change?"

_jcm, 2022, doi:10.3390/jcm11072058_

Round 1

Reviewer 1 Report

This paper reviews the impact of ambulatory care on a group of ACHD patients followed at a tertiary centre pre Covid and during a period of lockdown constraint.

It is an interesting and thought provoking contribution. Particularly attractive aspects were attempts to reduce bias in the patient sample, to include a 'typical' clinic population and present an objective look at the burden and benefit of standard outpatient ACHD practice including measures assessing impact on both patient and provider. Patients were surveyed by questionnaire (although detail of that survey and the response rate are not noted) and the institutions 'patient public involvement group' were also apparently very interested, although there is no detail as to whether they  contributed specific ideas or hypotheses.  It is of course difficult to ask those who don't attend why they are not there but just to note it is possibly of limited value to ask those who are in the waiting room to speculate on others who vote with their feet.

Some general assumptions were made (calculations on travel time as an example) but I do not believe these undermine the principles of the analysis.

There is presumably a degree of subjectivity to the assessment of whether specific clinical decisions were made at a given appointment although one hopes that the electronic medical record forces some commitment as to clinic outcome which would make this more transparent - it would be useful to understand more about this. One assumes that the authors noted "new findings" on ECG and or Echo when these were specifically noted by the clinician rather than reviewing those investigations or source data themselves?  

For a reader one question that arises with regard to new diagnostic findings would be "what would be the percentage that makes a test worthwhile". i confess to no specific familiarity with the literature on this. The point about newer methods of remote monitoring or use of biomarkers is well made and warrants further assessment. Many of us have had patients present with arrhythmia detected as an asymptomatic change in heart rate on a wearable device. 

In addition to surveillance I would argue that clinic visits also can usefully function to support patients with regard to ongoing education, reassurance, assessment of the psychosocial impact of chronic disease, discussion of the impact of chronic disease on life choices over time etc. They provide an opportunity to share  new clinical knowledge with the patient as this emerges. Traditional notions of healing might suggest that the presence of the clinician and patient in the same room and perhaps even "laying on of hands" might be of some health benefit. If the authors are suggesting a radical shake up of the current model it would be useful to understand how these various factors might weigh in the overall equation of the burden of a hospital attendance.

The study includes data on emergency presentations between clinic visits and notes that in many cases these were not predicted. It also notes that these presentations were often to non specialist hospitals - which is the reality of accessing urgent health care in many if not most urban settings. In the discussion the authors mention the role of primary care but it would be useful to understand their view of the role of secondary care for this group. If indeed acute presentations cannot be predicted with any certainty, is there a role, perhaps assisted by decision tools, for patients holding their own baseline data such as their ECG, for clear online guidelines for our secondary care centres, for even a patient-centric tool which looks at diagnosis and severity, predicts possible acute events and gives more specific management guidance.

It was quite startling living in a much more multicultural setting to see a clinic group of 100% 'white British' individuals. The authors do note this may limit the generalisability of their findings.  

Overall the tables and figures were fairly clear. Table 1 I found a little difficult to read- the combination of demographics and diagnoses, the line spacing and the table wrapping over a page break are a bit confusing.   The supplementary figures are interesting but need some further explanation

Reviewer 2 Report

This is a well-written retrospective study to explore how well outpatient clinic detects clinically relevant problems in adult CHD patients and consider the costs to the patient and health service provider in a tertiary center.

This paper gives new insight for outpatient care for adult CHD patients.

Because this paper was well written, I just suggest minor corrections below.

  1. In the 183rd line of 7th page, delete wide space before ’77 patients’.
  2. In Figure 2 legend, please delete 1 in the 1b).
  3. In Figure 3, please write full spelling of ‘DNA’ for clear understanding of readers.
